# ViTally Consistent: Scaling Biological Representation Learning for Cell Microscopy

## Abstract

Large-scale cell microscopy screens are used in drug discovery and molecular biology research to study the effects of millions of chemical and genetic perturbations on cells. To use these images in downstream analysis, we need models that can map each image into a feature space that represents diverse biological phenotypes *consistently*, in the sense that perturbations with similar biological effects have similar representations. In this work, we present the largest foundation model for cell microscopy data to our knowledge, a new 1.9 billion-parameter ViT-G/8 MAE trained on over 8 billion microscopy image crops. Compared to a previous published ViT-L/8 MAE, our new model achieves a 60% improvement in linear separability of genetic perturbations and obtains the best overall performance on whole-genome biological relationship recall and replicate consistency benchmarks. We also show these performance trends hold on a public benchmark for measuring compound activity against target genes. Beyond scaling, we developed two key methods that improve performance: (1) training on a curated and diverse dataset; and, (2) using biologically motivated linear probing tasks to search across each transformer block for the best candidate representation of whole-genome screens. We find that many self-supervised vision transformers, pretrained on either natural or microscopy images, yield significantly more biologically meaningful representations of microscopy images in their intermediate blocks than in their typically used final blocks, therefore enabling significant cost and energy savings when deploying these large models in real-world applications. More broadly, our approach and results provide insights toward a general strategy for successfully building foundation models for large-scale biological image data.

## 1 Introduction

Large-scale cell microscopy assays are used to discover previously unknown biological processes (Przybyla & Gilbert, 2022; Bock et al., 2022; Rood et al., 2024) and identify novel drug candidates and targets (Vincent et al., 2022). Labs are now able to achieve extremely high throughput by leveraging high content screening (HCS) systems that combine automated microscopy with robotic liquid handling (Boutros et al., 2015). Extracting meaningful features from microscopy images in large-scale screens has become increasingly difficult as this scale has increased. Public datasets like RxRx3 (Fay et al., 2023) and JUMP-CP (Chandrasekaran et al., 2023) now include millions of cellular images across 100,000s of unique chemical and genetic perturbations. In addition to limitations in expressiveness of the features that can be derived from them, traditional methods relying on customized pipelines for segmentation, feature extraction, and downstream analysis (Caicedo et al., 2017) struggle to handle this scale effectively (Chandrasekaran et al., 2021; Carpenter et al., 2006a).

The size and complexity of large-scale microscopy data demands image models that can extract rich biological features and do so consistently across experimental replicates, both of which are crucial for downstream biomedical applications. Rich, biologically meaningful representations reveal relationships between genes or compounds to drive the discovery of novel targets and drug candidates, while consistency in features extracted across batches and replicates ensures that findings are reproducible and reliable for therapeutic development.

Foundation models have been developed for representing high-dimensional unstructured biological data such as protein structures (Jumper et al., 2021) and transcriptomics (Hao et al., 2024), but the

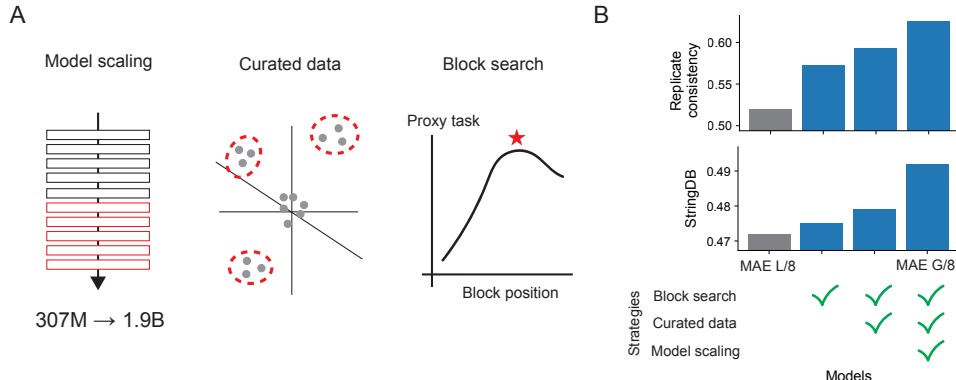

Figure 1: (A) Overview of performance gain from different MAE pretraining and inference strategies. (B) Example whole-genome results for **replicate consistency** and **biological relationship recall** on StringDB for models trained with different combinations of strategies, by model name and dataset (left to right):

scale and dimensionality of large-scale microscopy data present unique challenges for generating representations that are both biologically informative and consistent across replicates. HCS datasets are often confounded by complex noise known as batch effects (Caicedo et al., 2017), stemming from differences between experimental batches and biological variability. These batch effects – including natural variation in cell populations – obscure the biological effects of perturbations and make it challenging to isolate the specific effects of the perturbations applied (Yang et al., 2019). Overcoming these obstacles with a model capable of generating robust, biologically meaningful representations can empower HCS to systematically interrogate gene function and identify novel drug candidates (Rood et al., 2024).

State-of-the-art (SOTA) deep learning methods for microscopy leverage Vision Transformers (ViT) (Dosovitskiy et al., 2020) trained with self-supervised learning (SSL) techniques (Balestriero et al., 2023) to learn unbiased representations from large-scale screens (Doron et al., 2023; Kim et al., 2023; Bourriez et al., 2024). Recent studies have demonstrated that ViTs trained as Masked Autoencoders (MAEs) (He et al., 2022; Singh et al., 2023) can effectively scale beyond previous approaches and outperform various supervised and smaller SSL models in capturing biologically informative representations of cell images (Kraus et al., 2024). However, the level of consistency found in these representations across a large number of experimental replicates was not previously reported. Furthermore, compared to recent multi-billion parameter transformers developed for natural images (Dehghani et al., 2023) and natural language (Llama3, 2024), model scale in microscopy lags behind (Kraus et al., 2024; Chen et al., 2023a) despite the existence of massive datasets.

This work offers the following contributions:

- We demonstrate that training on a **curated microscopy dataset** of statistically significant positive samples, named Phenoprints-16M, improves both recall of known gene-gene relationships and consistency of embeddings for gene knockout perturbations (Figure 1A). We describe components of this curation strategy that can be generalized to other scientific datasets (§ 3.1).

- We present a **new foundation model, MAE-G/8**, a 1.86 billion parameter ViT-G/8 MAE trained on Phenoprints-16M over 48,000 H100 GPU hours on more than 8 billion samples from the curated dataset (Figure 1A, § 3.2).

- We propose a new set of **biological linear probing tasks** to evaluate representations learned by intermediate ViTs blocks for microscopy data (§ 4). Performance on these linear probing tasks are strongly correlated with performance on important whole-genome scale evaluation metrics while requiring significantly less resources to compute (Figure 4). These trends hold when evaluating on a public benchmarking dataset called RxRx3-core, where our new ViT-G/8 obtains the best average precision for zero-shot cosine similarity prediction of target gene activity across thousands of compounds (§ 6).

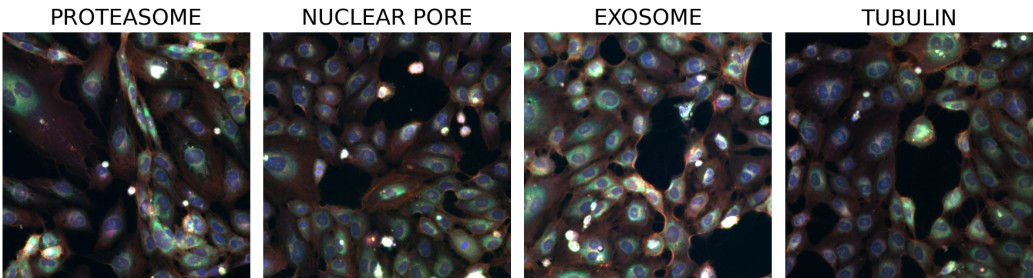

PROTEASOME NUCLEAR PORE EXOSOME TUBULIN

Figure 2: Samples for subset of groups in Anax 40-class functional gene group classification task.

- We find across SSL ViTs trained on microscopy or natural images that **using intermediate layers leads to better performance** on these downstream whole-genome benchmarks at a lower computational inference cost. By taking advantage of our linear probing proxy task, we are able to cheaply find the best performing intermediate block (Eq. 1).
- Lastly, our results indicate that the scaling properties first identified by Kraus et al. (2023) extend to the **multi-billion parameter model regime** across a wide variety of newly examined biologically-motivated benchmark tasks for this data modality where large models have been under-studied in comparison to natural images and natural language.

## 2 RELATED WORK

**Evaluating Representations for Drug Discovery.**  Evaluating the quality of biological representation learning methods for drug discovery remains challenging, as ground truth data is sparse, noisy, biased to well-studied diseases and pathways, and poorly annotated. Metrics have been proposed that use mean average precision (Kalinin et al., 2024) or AUC ROC (Sivanandan et al., 2023) to assesses how similar related samples are represented, including replicates of the same perturbation or different perturbations with similar annotated biological activities. Recently, Celik et al. (2024) introduced terminology for describing perturbative "maps of biology", in which representations of perturbations in HCS data can be placed in unified, relatable embedding spaces allowing for the generation of genome-scale sets of pairwise comparisons. Here we leverage the *biological relationship recall* benchmark proposed by Celik et al. (2024), which assess how well known relationships between pairs of perturbations are recalled among the most similar or dissimilar embeddings. Computing reliable versions of these relationship benchmarks with HCS data is particularly expensive as they require genome-wide embeddings to be inferred for hundreds of millions of image crops from the genome-wide RxRx3 microscopy screen (Fay et al., 2023).

**Dataset Curation for Foundation Models.**  Dataset curation is crucial for enhancing the efficiency of foundation models, especially in large-scale contexts. Usual approaches to dataset construction are inspired by the image retrieval community (Weinzaepfel et al., 2022; Radenović et al., 2018; Berman et al., 2019). Existing methods often utilize pre-trained models for filtering and pruning, such as vision-language models to discard irrelevant pairs (Schuhmann et al., 2021), semantic deduplication to remove redundancy (Abbas et al., 2023), and prototypicality-based approaches to retain representative data (Sorscher et al., 2022). However, these techniques are less effective for HCS, where redundancy, variability, and subtle morphological differences make conventional filtering challenging. Our work addresses these limitations by building on Celik et al. (2024)'s *perturbation consistency* framework to curate a balanced dataset of images across semantic classes, which is vital for effective learning under the masked objectives (Zhang et al., 2022).

**Layer-wise Analysis of Deep Neural networks.**  Recent work suggests that intermediate layers (or, blocks) in large ViTs may achieve superior performance on certain linear probing tasks compared to the final encoder layer (Evci et al., 2022; Dehghani et al., 2023). Alkin et al. (2024) reported that intermediate layers in large MAE-ViTs (ViT-L, ViT-H) have superior ImageNet-1K $k$-NN accuracy, likely because later encoder layers become more optimized for the reconstruction task.

# 3 Vision Transformers for Microscopy Images

We train and evaluate various vision transformers (ViTs, Table 4) as encoders to extract feature embeddings from $256 \times 256 \times 6$ (HxWxC) microscopy image crops (Figure 2).

## 3.1 Training Dataset Curation

Many academic and industry labs have adopted the Cell Painting imaging protocol (Bray et al., 2016), which multiplexes fluorescent dyes to reveal eight broadly relevant cellular components. The datasets used here contain a six-channel implementation of Cell Painting (Figure 2), as well as brightfield images, spanning 100,000s of chemical and genetic perturbations applied to dozens of cell types (Kraus et al., 2024). In these datasets, cells that look like unperturbed cells tend to be very over-represented because many perturbations do no induce a morphological change. Some morphological changes are also far more common (e.g. many perturbations will kill cells, resulting in a relatively high proportion of dead cell morphological phenotype). This results in significant imbalance in the morphological phenotypes that the models learn to reconstruct.

To address this, we constructed an aggressively curated training dataset (§ A.1). To learn an initial representation, we began by reproducing the MAE-L/8 model of Kraus et al. (2024) on a dataset of similar size consisting of 93 million HCS images. Using this representation, we first filtered perturbations that did not induce consistent morphological changes to cells. To perform this filtering, we utilized Celik et al. (2024)'s non-parametric perturbation consistency test (§ A.3) after correcting for batch effects using Typical Variation Normalization (Ando et al., 2017; Kraus et al., 2024). This test was applied within each experiment for computational efficiency, and we restricted the analysis to wells containing single perturbations. This consistency was computed for CRISPR guides, siRNAs, and particular concentrations of small molecules across replicates of the same perturbation. P-values were computed for each gene and each (perturbation, concentration) pair. When multiple experiments existed for the same condition, we combined p-values using the Cauchy Combination test (Liu & Xie, 2018).

We repeated this procedure with a weakly supervised learning (WSL) model trained on RxRx1 (Sypetkowski et al., 2023) and filtered to perturbations where any condition had a p-value $< 0.01$ in either the MAE-L/8 or WSL model. This process reduced our original dataset of 93M samples to 16M, which we refer to as Phenoprints-16M. While some redundancy remains when distinct perturbations have the same effect, the proportion of samples with that differ from negative controls increased substantially with little decrease in overall diversity. We believe that iteratively repeating this process with the best models from previous iterations to guide data selection for subsequent models may be a viable strategy.

## 3.2 Models

**Baselines.** We compare to several non-finetuned baseline ViT image encoders: three different Dino-v2 backbones (Oquab et al., 2024) (with 4 register tokens (Darcet et al., 2024)) trained on a curated non-biological natural image dataset; a weakly supervised (WSL) classifier ViT-L/16 trained on Imagenet-21k (Ridnik et al., 2021); a MAE ViT-L/16 trained on Imagenet-21k (He et al., 2022); and an untrained ViT-S/16. We found that channel-wise self-standardization worked best as the image normalization preprocessing for these baselines, and that the class token was slightly better than the global pool of the patch tokens (except for MAE). Convolutional weights in the patch embedding layer were repeated to embed 6 channel images when using models trained on RGB datasets (Wightman, 2019).

**Prior work.** Our primary point of comparison is with respect to the best pretrained foundation model presented by Kraus et al. (2024), the MAE-ViT-L/8+ trained on RPI-93M. This MAE-L/8 was trained for approximately 40 epochs, learning from over 3.5 billion image crops, using the L2 mean squared error loss function plus an additional Fourier domain reconstruction loss term.

**CA-MAE-S/16 trained on RxRx3.** We trained a new channel-agnostic MAE (Kraus et al., 2024) ViT-S/16 on the RxRx3 dataset (Fay et al., 2023) for 100 epochs. Channel-agnostic ViTs tokenize each image channel separately with shared patch embedding weights and leverage the dynamic

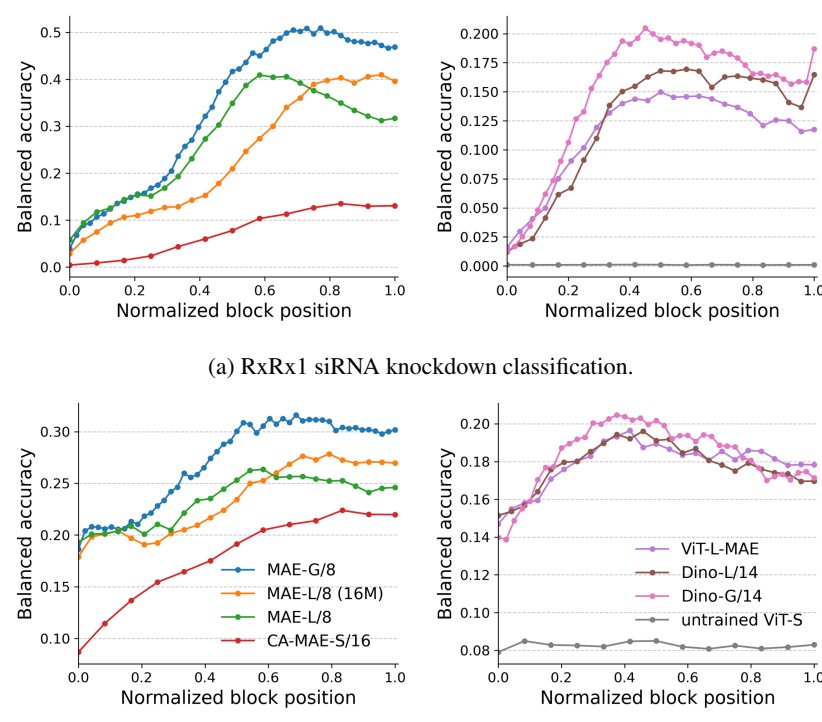

(a) RxRx1 siRNA knockdown classification.

(b) Anax functional gene group classification.

Figure 3: Block-wise validation set **linear probe results** comparing ViT models pretrained on cell microscopy images (left) versus natural images (right). (a) 1139-class RxRx1 SiRNA knockdown classification (Sypetkowski et al., 2023); (b) 40-class Anax functional gene group classification on HUVEC cell images from RxRx3 CRISPR knockouts (Fay et al., 2023).

sequence length of transformers with repeated positional encodings to train ViTs that can process images with varying numbers of channels (Bao et al., 2024; Bourriez et al., 2024; Kraus et al., 2024). Kraus et al. (2024) demonstrate that the large MAEs with 8x8 patch size perform either better or the same as the 16x16 channel-agnostic variants for consistently 6-channel data, so we opted to train standard MAEs for the following two new models since they require fewer tokens at inference time.

**MAE-L/8 trained on Phenoprints-16M.** Holding the model backbone constant compared to the MAE-ViT-L/8 by Kraus et al. (2024), we assess the impact of our curated dataset in contrast to the 93M dataset by training a new ViT-L/8 MAE for 500 epochs on Phenoprints-16M.

**MAE-G/8 trained on Phenoprints-16M.** Holding the dataset constant compared to MAE-L/8 above, we assess the impact of increased model scale in terms of parameters by training a new ViT-Gigantic MAE with nearly 1.9 billion parameters for 500 epochs on Phenoprints-16M. Training this model required 256 H100 GPUs running in parallel for over 1 week. See § A.2 for other hyperparameter settings we used for model training.

## 4 LINEAR PROBING REPRESENTATION LEARNING ACROSS VIT BLOCKS

We improve the quality of our learned image representations by leveraging previous findings that suggest intermediate blocks within an encoder can provide better representation compared to the final block (Alkin et al., 2024). Unfortunately, it is infeasible to search for the best block by simply performing whole-genome evaluation on each block of a large model because the evaluation is extremely time-consuming and resource intensive. For example, evaluating the final block of MAE-G/8 required 4,000 L4 GPU hours just for inference (§ 5). We demonstrate that using block-wise linear probes provides insights into the quality of biological features extracted by these models

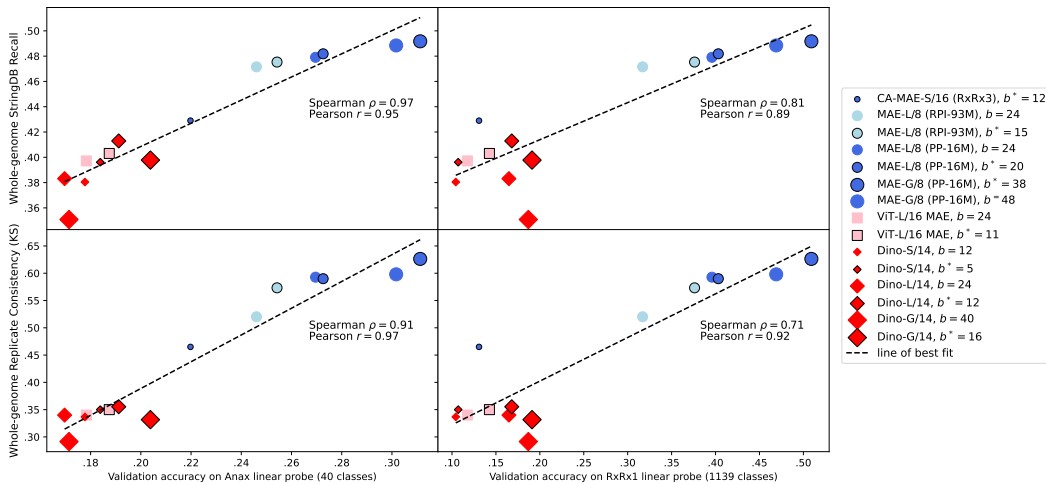

Figure 4: **Correlations** between validation set linear probing (Figure 3) on Anax and RxRx1 for best and last blocks (Eq. 1) compared to downstream whole-genome benchmarks (Table 1) for biological relationship recall on StringDB at 0.05-0.95 threshold and replicate consistency KS statistic. Models with **bold** borders are *trimmed*, red are natural image baseline models and blue are trained on microscopy.

in their intermediate blocks, allowing us to trim the model to an earlier block to both reduce inference costs and improve representation quality.

Our block-wise search consists of training a logistic regression model (linear probe) on the output features of each transformer block to predict either the gene that was perturbed or the functional group that the gene belongs to, and test performance on held-out experiments (§ A.4). We define the optimal block $b^*$ for a probing task as the block whose output features achieve the highest test balanced accuracy when trained on the probing task, across all $N$ blocks of the encoder,

$$b^* = \underset{b \in \{1,2,...,N\}}{\arg\max} \ \text{BalancedAccuracy}(\mathbf{z}^{(b)}), \tag{1}$$

where $\mathbf{z}^{(b)}$ are output features from block $b$ of a ViT. Performance on our linear probing tasks can be viewed as a measure of linear separability of a feature space across experimental batches.

**RxRx1 1139-class siRNA genetic perturbation classification.** We expect high quality representations of cell images to generate similar embeddings for cells with the same perturbation, hence a simple linear probe should be able to predict gene perturbation from these representation reasonably well. We train linear probes on the publicly-available RxRx1 dataset Sypetkowski et al. (2023) which consists of 125,510 high-resolution fluorescence microscopy images of human cells under 1,138 siRNA-induced gene knockdowns (plus unperturbed controls) across four cell types (HepG2, HUVEC, U2OS, RPE). These gene knockdowns produce strong phenotypes which makes the prediction task more feasible.

We found that, for MAE-G/8 , the best features came from intermediate block $b^* = 38$ (out of $48$) of the encoder, achieving a balanced accuracy $(0.51)$ that is $8.5\%$ greater compared to its final block's output features (Figure 3a, left). Additionally, these features achieved $60\%$ greater accuracy than the typically used final block of MAE-L/8+ (Kraus et al., 2024). We observed similar trends for ViT models pretrained on natural images. For example, DINO-G/14 and ViT-L/16 MAE trained on non-biological natural image data have their best features at blocks that are positioned within the first half of the encoder. For ViT-L/16 MAE, the performance of the best block is $27\%$ higher compared to its final block output features that are typically used for downstream tasks. The higher performance observed for intermediate blocks does not appear to be an intrinsic feature of the ViT architecture as an untrained ViT did not exhibit such a parabolic trend (Figure 3a, right).

**Anax 40-class functional gene group classification.** Biologically meaningful representation of microscopy images of genetically perturbed cells should capture functional relationships between genes, hence a simple linear probe should be able to predict functional gene groups when trained on these representations. We curated a small subset of 80,000 wells from RxRx3 (Fay et al., 2023) to evaluate linear probes on functional group prediction. We also evaluated similar whole genome knockout screens with ARPE-19 and an additional population of HUVEC cells with soluble TNF-$\alpha$ added to all wells. We manually curated Anax, a set of 40 functionally-diverse gene groups containing 348 genes, with details provided in (§ A.8). Examples of groups include major protein complexes (e.g. proteasome, ribosome-small/large), metabolic pathways (e.g. Krebs cycle) and signaling pathways (e.g. calcium signaling) (Figure 2). These groups span broad biological processes that are conserved across cell types – linear separability of these groups would likely indicate that representations are biologically meaningful regardless of cell type.

As shown in Figure 3b, MAE-G/8 significantly outperforms other models in Anax group linear probe classification. The best representations once again are obtained from an intermediate block, achieving a balanced accuracy (0.32) that is $5\%$ greater compared to its final block. We observed similar trends for ViT models pretrained on natural images and representations computed from microscopy images of other cell types/conditions (§ A.5, Figure 7).

In Figure 4, we observe that performance on this novel linear probing task correlates strongly with downstream whole-genome benchmarks across all models (Table 1), whether they are trained on microscopy data or natural images, achieving an overall rank correlation $\rho = 0.97$ with whole-genome StringDB recall and $\rho = 0.91$ with whole-genome replicate consistency. This strong correlation is crucial as it allows us to trim our model to the block with the best linear probe performance as a way to improve the quality of our representations for the whole-genome (Table 1).

## 5 WHOLE-GENOME BENCHMARKING

Table 1 presents our benchmarks computed across the whole-genome. These evaluate the genomic representations obtained for each model by aggregating millions of embeddings of cell images spanning >100,000 of genetic knockout perturbations (17,063 genes $\times$ 6 single guide RNAs each) on HUVEC cells from RxRx3 (Fay et al., 2023). Computing these benchmarks for HCS screens typically requires inferring 140 million crops from the genome-wide RxRx3 microscopy screen (Kraus et al., 2023) (64 tiled crops per each of the 2.2 million wells), but, to reduce compute costs, we discard the outer ring of crops, leaving the 36 center non-edge crops for each well. This requires 80 million forward passes to comprehensively evaluate a new encoder. After inference, we use typical variation normalization (Ando et al., 2017) and chromosome arm bias correction (Lazar et al., 2024) to post-process the embeddings and aggregate them to the gene-level.

We present the multivariate **biological relationship recall** benchmarks proposed by Celik et al. (2024) and originally evaluated for MAEs by Kraus et al. (2023; 2024). These metrics evaluate how many annotated pair-wise relationships are recalled from public databases (CORUM, hu.MAP, Reactome-PPI, StringDB) in the extremities of a ranked list of cosine similarities of all pair-wise post-processed embeddings (details in § A.6). To ensure embeddings represent technical replicates of perturbations consistently, we also evaluate model performance on **replicate consistency** based on the experimental design used in the RxRx3 dataset. Specifically, we compare the similarity of the embedding for corresponding wells across different experiments via a non-parametric statistical test. The test statistic measures the difference between the perturbation replicates' similarity distribution and an empirical null distribution, with larger values indicating greater consistency (details in § A.7).

In order to compare models, we summarize the resulting statistics over all technical replicates in RxRx3 by taking their median, as reported in columns KS and CVM in Table 1. Even the smallest CA-MAE-S/16 trained on microscopy data outperforms all of the large baselines trained on natural images. Furthermore, training on the Phenoprints-16M dataset improves the performance of the MAEs, and MAE-G/8 achieves the best overall performance. Compared to the best published result for whole-genome benchmarks (MAE-L/8 trained on RPI-93M (Kraus et al., 2023)), MAE-G/8 obtains a 48% improvement in replicate consistency CVM (12.3→18.2) and 4.3% improvement in StringDB recall (.472→.492). Using linear probes to select block $b^* = 15$ (Equation 1) for that MAE-L/8, our improvement changes to 20% in CVM and 3.5% in StringDB recall.

| Model backbone | $b$ | CORUM | hu.MAP | React. | StringDB | KS | CVM |
|---|---|---|---|---|---|---|---|
| **Baseline ViTs** | | | | | | | |
| ViT-S/16, Untrained | 12 | .452 | .343 | .205 | .359 | .30 | 4.3 |
| ViT-L/16, ImageNet WSL | 24 | .518 | .351 | .210 | .394 | .34 | 5.5 |
| ViT-L/16, ImageNet MAE | 24 | .526 | .355 | .215 | .397 | .34 | 5.1 |
| *trimmed* | 11 | .532 | .359 | .218 | .403 | .35 | 5.8 |
| **Baseline Dino ViTs** | | | | | | | |
| ViT-S/14, Dino-V2 | 12 | .484 | .345 | .203 | .380 | .34 | 5.6 |
| *trimmed* | 5 | .514 | .359 | .213 | .396 | .35 | *6.0* |
| ViT-L/14, Dino-V2 | 24 | .492 | .339 | .210 | .383 | .34 | 5.3 |
| *trimmed* | 12 | *.549* | *.367* | *.220* | *.413* | *.36* | 5.9 |
| ViT-G/14, Dino-V2 | 40 | .442 | .312 | .199 | .351 | .29 | 3.8 |
| *trimmed* | 16 | .529 | .354 | *.220* | .398 | .33 | 5.2 |
| **MAEs for microscopy** | | | | | | | |
| CA-MAE-S/16 , RxRx3 | 12 | .549 | .374 | .229 | .429 | .47 | 10.4 |
| MAE-L/8 , RPI-93M | 24 | .609 | .434 | .251 | .472 | .52 | 12.3 |
| *trimmed* | 15 | .602 | .427 | .255 | .475 | .57 | 15.2 |
| MAE-L/8 , PP-16M | 24 | .600 | .432 | .255 | .479 | .59 | 16.2 |
| *trimmed* | 20 | .600 | .435 | .260 | .482 | .59 | 16.2 |
| MAE-G/8 , PP-16M | 48 | **.621** | **.438** | **.263** | .488 | .60 | 16.4 |
| *trimmed* | 38 | .615 | .437 | **.263** | **.492** | **.63** | **18.2** |

Table 1: Multivariate **known biological relationship recall** and univariate **replicate consistency** benchmarks by model, encoding block $b$, benchmark database (CORUM, hu.MAP, Reactome-PPI, and StringDB), and consistency test statistics (KS and CVM). The *trimmed* models used linear probes to select an earlier block as the feature encoder (Fig. 3). Results are computed over all >17,000 whole-genome CRISPR knockout perturbation images in RxRx3, after applying TVN and chromosome arm bias correction. Over all benchmarks, higher is better, and best overall result is in **bold**, and best result among baselines is in *italics*. For relationship recall we report the mean @ 0.05-0.95 cosine threshold over 3 random seeds sampling the null distribution for each benchmark run (standard deviation for each result is $\leq \pm.0015$).

| Model backbone | $b$ | Pretraining data | CORUM | hu.MAP | Reactome | StringDB |
|---|---|---|---|---|---|---|
| CellProfiler | - | N/A | .219 | .184 | .131 | .191 |
| CA-MAE-S/16 | 12 | RxRx3 | .233 | .199 | .154 | .214 |
| MAE-L/8 | 24 | RPI-93M | .248 | .208 | .160 | .226 |
| MAE-G/8 | 38 | Phenoprints-16M | **.264** | **.215** | **.165** | **.235** |

Table 2: Biological relationship recall benchmarks at 0.05-0.95 cosine threshold on public JUMP-CP image data (Chandrasekaran et al., 2023) generated by completely different labs and assay protocols compared to the data used for pretraining. Each result has a standard deviation $\leq \pm.0023$, and spans nearly 8,000 gene-knockouts and are computed after applying PCA with center-scaling for embedding post-processing alignment.

Similarly, linear probing to select optimal ViT blocks led to significant improvements even when applied to Dino-V2 based models pretrained on natural images. Dino-V2 ViT-G obtains a nearly 20% improvement in recall on CORUM (.44→.53) by using the embeddings extracted at $b^* = 16$ (chosen by linear probes) rather than the final embedding from $b = 40$ (which performs worse than a random untrained ViT-S). Dino-V2 ViT-S also observes improvements by using $b^* = 5$ rather than $b = 12$ and outperforms Dino-V2 ViT-G in replicate consistency. We also attempted to train Dino-V2 on microscopy data, but preliminary results were worse than CA-MAE-S/16 (§ A.9).

Additional results indicate that these MAEs effectively generalize to novel microscopy data generated in different labs with different assays (Table 2, described in § A.11), and that scaling trends for performance on biologically relevant tasks continues from MAE-L/8 to MAE-G/8 (§ A.10).

| Model | Avg. prec. | Comp. z-score ↑ | Energy dist. | Energy z-score ↑ |
|---|---|---|---|---|
| Random baseline | $0.222 \pm .007$ | 0.00 | $0.728 \pm 0.05$ | 0.00 |
| CA-MAE-S/16, RxRx3 | $0.273 \pm .016$ | 2.90 | $3.319 \pm 0.83$ | 3.10 |
| MAE-L/8, RPI-93M | $0.290 \pm .017$ | 3.77 | $4.856 \pm 1.25$ | 3.30 |
| *trimmed* | $0.299 \pm .016$ | 4.49 | $4.514 \pm 1.16$ | 3.26 |
| MAE-G/8, PP-16M | $0.302 \pm .015$ | 4.79 | $6.053 \pm 1.47$ | 3.63 |
| *trimmed* | $\mathbf{0.309} \pm .015$ | **5.38** | $\mathbf{6.586} \pm 1.52$ | **3.85** |

Table 3: Performance on the public **RxRx3-core compound-gene benchmark** and **RxRx3-core perturbation magnitude benchmark**, measuring mean average precision ($\pm$ STD over 100 random seeds benchmarking with different negative samples) in predicting compound activity against target genes, and mean energy distance ($\pm$ MAD over all perturbations) separating perturbation embeddings from controls with corresponding z-scores of improvement over a random baseline.

## 6    RxRx3-core benchmarking

RxRx3-core[1] is a publicly available benchmarking dataset for assessing biological capabilities of computer vision models. RxRx3-core includes labeled images (compressed to JPEG-2000) of 735 genetic knockouts and 1,674 small-molecule perturbations across eight concentrations drawn from 222,601 wells ($512 \times 512 \times 6$ pixel center-crops) drawn from the larger RxRx3 dataset.

We evaluate a random embedding baseline, the CA-MAE-S/16 model, the MAE-L/8 model from previous work (Kraus et al., 2023), and the MAE-G/8. We evaluate both the trimmed and full-length version of the latter two models to determine the impact of our model-trimming strategy in this context. To evaluate each model, we first inference all $222,601 \times 4$ crops and then average the 4 embeddings to the well-level. Then, to perform standard batch correction alignment, we use the "EMPTY", unperturbed wells as our control population. We fit PCA on those control embeddings, use it transform the rest, and then fit a separate standard-scaler on each batch's controls to transform the rest. This simplified alignment strategy empirically performed better TVN only on this dataset.

We present results for the benchmark measuring zero-shot prediction of compound-gene activity using cosine similarities between embeddings (Figure 5). This measures, for each compound, whether the cosine similarities from a model's embeddings correctly rank the compound's known target genes higher than a randomly sampled set of other genes from a ground truth dataset. Table 3 provides exact values along the *max* axis, which captures the strongest potential interaction regardless of concentration. The relative ranking of model performance, holds as expected from the results in § 4 and § 5, and trimming benefits both MAE-L/8 and MAE-G/8, with MAE-G/8 offering a 42% (3.77 $\rightarrow$ 5.38) improvement over the method from previous work in predicting compound-gene activity.

In Table 3 we also present the results from the perturbation magnitude benchmark (Celik et al., 2024), which measures the energy distance between perturbation embeddings and control embeddings (visualized in Figure 6). Unlike on the rest of the benchmarks, the MAE-L/8 trained on RPI-93M does not benefit from trimming here. But, MAE-G/8 obtains the best performance overall and improves when trimmed to the best layer as detected by our linear probes.

## 7    Discussion and Conclusions

This work demonstrates that: (1) within the context of biological imaging, trimming many ViTs to an earlier block leads to stronger biological linearity and improved performance on downstream tasks in addition to cheaper inference costs (Figure 3); (2) linear probing performance on a subset of genetic perturbations correlates strongly with downstream performance on whole-genome benchmarks and can be used to optimize which block is selected for representing the whole-genome (Figure 4); (3) the most scaled model, MAE-G/8 , obtains the overall best performance across all benchmarks and linear probes, providing further evidence for the scaling hypothesis in biological image data (Table 1, Table 3, § A.10). This demonstrates that intentionally scaling training compute and parameters of SSL models for microscopy can benefit a wide variety of biologically relevant tasks.

---

[1]`huggingface.co/datasets/recursionpharma/rxrx3-core`

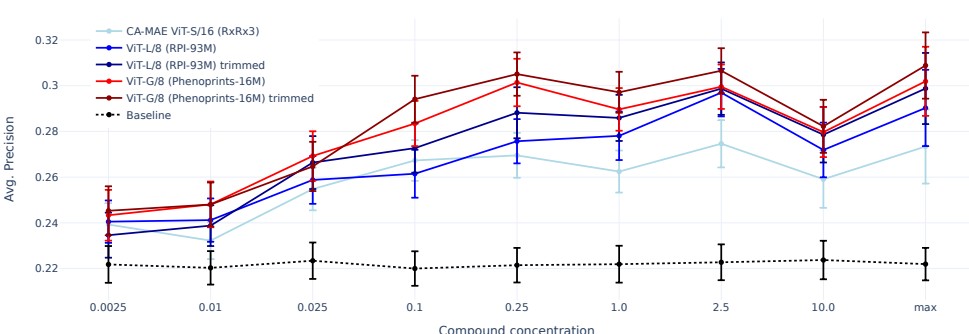

Figure 5: Mean average precision performance on RxRx3-core public benchmark in predicting compound activity against annotated gene targets, across all compound concentrations with error bars for 100 runs of the benchmark with different random seeds (Table 3).

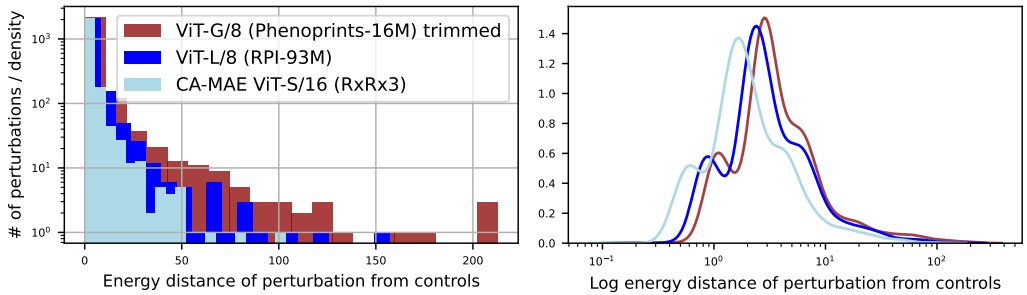

Figure 6: Distribution of perturbation magnitudes for different models as measured by energy distance between model embeddings of perturbations versus controls on RxRx3-core (Table 3).

More broadly, this work proposes a reusable recipe for training and extracting optimal representations from fully self-supervised models trained on experimental data. The pattern we use can be applied to other domains that contain data from repeated experiments but without accurate ground truth labels. Specifically, we recommend: (1) curating the training set by identifying diverse sets of samples that are represented consistently, e.g., by using a pre-existing model to select such samples; (2) training a scaled transformer-based model using a self-supervised learning technique, such as masked autoencoding; and, (3) evaluating the performance of the trained transformer at every block to identify the optimal layer for representing the data.

## LIMITATIONS AND REPRODUCIBILITY

In this work, we evaluated baselines and new linearly probed MAE models trained on a specially curated microscopy dataset. Our preliminary attempts to train ViTs with DINO on this microscopy data encountered suboptimal performance (§ A.9). Consequently, we allocated our time and compute budget to investigate scaling MAE to ViT-G/8. We recognize the potential for other SSL training regimes and fine-tuning strategies (Singh et al., 2023; Lehner et al., 2024; Hondru et al., 2024; Khan & Fang, 2024; Alkin et al., 2024) oriented for microscopy data to lead to future improvements on these tasks. With this work, we can publicly release the training, inference, reconstruction visualization, and benchmarking code[2], with the full weights[3] for CA-MAE ViT-S/16.

---

[2]github.com/[redacted]
[3]huggingface.co/[redacted]

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

Table 4: Overview of vision transformer (ViT) encoders used and evaluated in this work.

| Model Name | Parameters | Blocks | Model Dim | Pretraining Data |
|---|---|---|---|---|
| **Baselines** | | | | |
| Untrained ViT-S/16 | 25M | 12 | 384 | N/A |
| Dino-V2 ViT-S/14 | 25M | 12 | 384 | Natural images |
| Dino-V2 ViT-L/14 | 307M | 24 | 1024 | Natural images |
| Dino-V2 ViT-G/14 | 1,100M | 40 | 1536 | Natural images |
| ViT-L/16 WSL | 307M | 24 | 1024 | Imagenet-21k |
| ViT-L/16 MAE | 307M | 24 | 1024 | Imagenet-21k |
| **MAEs for microscopy** | | | | |
| CA-MAE-S/16 | 25M | 12 | 384 | RxRx3 |
| MAE-L/8 | 307M | 24 | 1024 | RPI-93M |
| MAE-L/8 | 307M | 24 | 1024 | Phenoprints-16M |
| MAE-G/8 | 1,860M | 48 | 1664 | Phenoprints-16M |

Samuel J. Yang, Scott L. Lipnick, Nina R. Makhortova, Subhashini Venugopalan, Minjie Fan, Zan Armstrong, Thorsten M. Schlaeger, Liyong Deng, Wendy K. Chung, Liadan O'Callaghan, Anton Geraschenko, Dosh Whye, Marc Berndl, Jon Hazard, Brian Williams, Arunachalam Narayanaswamy, D. Michael Ando, Philip Nelson, and Lee L. Rubin. Applying Deep Neural Network Analysis to High-Content Image-Based Assays. *Slas Discovery*, 24(8):829–841, 2019. ISSN 2472-5552. doi: 10.1177/2472555219857715.

Xiaohua Zhai, Alexander Kolesnikov, Neil Houlsby, and Lucas Beyer. Scaling vision transformers. In *Proceedings of the IEEE/CVF Conference on Computer Vision and Pattern Recognition*, pp. 12104–12113, 2022.

Qi Zhang, Yifei Wang, and Yisen Wang. How mask matters: Towards theoretical understandings of masked autoencoders. In *NeurIPS*, 2022.

# A APPENDIX

## A.1 TRAINING DATASET CURATION DETAILS

In order to produce Phenoprint-16M, we curated 93M using the following steps:

1. Filtering out data that did not pass data quality filters related to the focus of the image, quantity of dead cells, assay conditions, and presence of strong anomalous imaging artifacts.

2. Filtering out data with missing information about the perturbations applied, data with more than 3 perturbations applied, and data of unusual size (in the image dimension or number of channels).

3. Filtering out perturbation conditions that had been in less than 3 distinct experiments or 20 distinct wells so as to capture a variety of batch effects and have a broad sample of positives per class.

4. Under-sampling perturbation conditions that were clearly over-represented in the dataset. Our experiment designs contain positive controls, negative controls, and wells without perturbation within each experiment. At this step, we keep 10% of positive controls and wells without any perturbation, 30% of negative controls, and all other perturbation conditions.

5. Filtering out wells where none of the perturbation conditions had a phenoprint (§A.3) (across different map types) in any experiment it had been run in.

## A.2 TRAINING HYPERPARAMETERS

Table 5 provides the hyperparameters used for training the new vision transformers presented in this work. Each model was trained using a 75% mask ratio and the standard decoder architecture for

Table 5: Training hyperparameters for the new models presented in this work. Each used a one-cycle cosine learning rate decay schedule with 10% warm-up using the Lion optimizer from Chen et al. (2023b) with betas (0.9, 0.95) and weight decay of 0.05, with additional ViT settings such as LayerScale as proposed by Dehghani et al. (2023). *Note that MAE-G/8 had multiple restarts during training due to challenges associated with massive model training on large-scale shared distributed compute clusters.

| Hyperparameter | CA-MAE-S/16 | MAE-L/8 | MAE-G/8 |
|---|---|---|---|
| Vision transformer backbone | ViT-S | ViT-L | ViT-G (Zhai et al., 2022) |
| Pretraining Data | RxRx3 | Phenoprints-16M | Phenoprints-16M |
| Training epochs | 100 | 500 | 500* |
| Learning rate | 1e-4 | 3e-5 | 3e-5 |
| Global batch size | 2048 | 16384 | 8192 |
| Stochastic depth | 0.1 | 0.3 | 0.6 |
| # GPUs | 16 A100s | 128 H100s | 256 H100s |
| # GPU-hours | 400 | 15,360 | 48,000 |

MAEs (He et al., 2022). Each model was trained with the standard L2 MAE loss and the Fourier-space loss function implemented by Kraus et al. (2024) with a weight of $\alpha = 0.01$. We note, however, that the details presented by Kraus et al. (2024) do not precisely correspond with the implementation provided in their Github repository; when reshaping the tokens to a shape compatible with the 2D Fourier transform, the permute operation resulted in adjacent pixels being from different channels of the input, resulting in the high frequency components of the loss being a function of the relationships between input channels. An initial investigation with a ViT-L/8 showed that changing the implementation to the one described in the paper did not dramatically change probing results. As such, we used the implementation as-is and leave additional analysis of loss function design for MAEs to future work.

### A.3 PERTURBATION CONSISTENCY

In order to assess the consistency of the induced morphology on the cells by the perturbations, we used a non-parametric perturbation consistency test similar to the one introduced in Celik et al. (2024). Let $x_{g,1}, x_{g,2}, \cdots, x_{g,n}$ be the embeddings for replicates of perturbation $x_g$ on experiment (batch) $e$. As the test statistic for perturbation consistency, $\bar{s}_g^e$ is defined as the mean of the cosine similarities across all pairs of replicates of $x_g$.

$$\bar{s}_g^e = \frac{1}{n^2} \sum_{i=1}^{n} \sum_{j=1}^{n} \frac{\langle x_{g,i}, x_{g,j} \rangle}{||x_{g,i}|| ||x_{g,j}||}. \tag{2}$$

where $\langle . \rangle$ and $||.||$ denote dot product and $L_2$ norm.

Statistical significance of $\bar{s}_g^e$ is assessed using a permutation test comparing it against an empirical null distribution generated using the same statistic for a set of randomly selected perturbations in experiment $e$, $\{\bar{s}_1', \cdots, \bar{s}_K'\}$. The p-value for $\bar{s}_g^e$ is computed as follows

$$p_g = \frac{\max\left\{ \#\{\bar{s}_k' \geq \bar{s}_g^e\}, 1 \right\}}{K}. \tag{3}$$

When multiple experiments existed for the same perturbation, we combined p-values using the Cauchy Combination test (Liu & Xie, 2018).

### A.4 TRAINING LINEAR PROBES

In this section, we provide details about the training process and preprocessing steps used in our logistic regression models. These models were trained on output features derived from various Vision Transformer (ViT) blocks.

The data was split by experiments, ensuring that the test data originated from experiments distinct from those used for training. This approach helps to validate the generalization performance of our models across different experimental conditions.

For both RxRx1 gene prediction and Anax group prediction, we apply `StandardScaler` from the scikit-learn library as the only preprocessing step to standardize the features prior to training linear probes. `StandardScaler` transformation was fitted on data from the train split. We trained the logistic regression models using scikit-learn's `LogisticRegression` class. The following parameters and settings were used during model optimization:

- Solver: lbfgs
- Maximum Iterations: 2000
- Class Weight: balanced

For RxRx1 gene prediction, we trained logistic regression models to predict one of 1139 possible perturbation labels (1138 genetic perturbation and non-perturbed control). For Anax group prediction, we trained logistic regression models to predict one of 40 possible function group labels (§ A.8). We report the balanced test accuracy as the main evaluation metric for all linear probing experiments.

### A.5 ANAX CLASSIFICATION FOR OTHER CELL LINES/TREATMENT CONDITIONS: ARPE19 AND HUVEC WITH TNF-ALPHA BACKGROUND

We performed linear probing on imaging data obtained for a retinal pigment epithelia (RPE) cell line, ARPE19, and HUVEC cells treated with an inflammatory cytokine, TNF$\alpha$. We similarly observed that intermediate blocks often have the most linearly separate features compared to the final block.

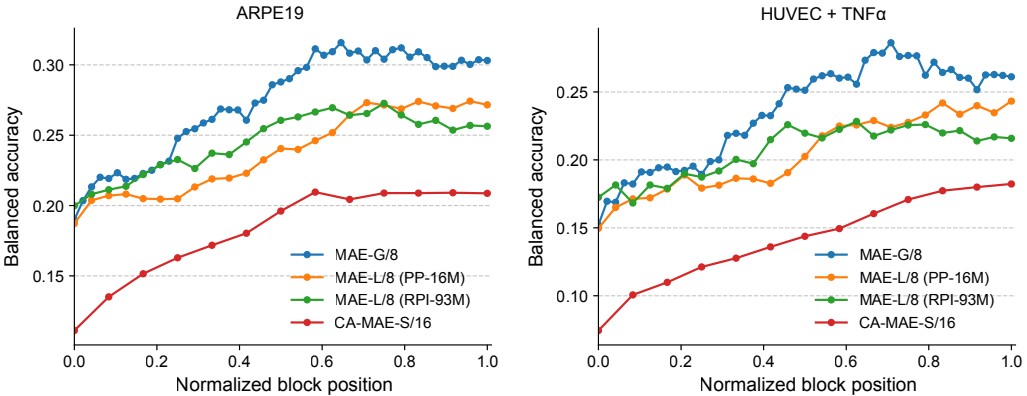

Figure 7: Layerwise validation set linear probe performance on Anax functional gene group classification beyond RxRx3: CRISPR knockouts in the ARPE-19 immortalized epithelial cell-line (left), and in HUVEC cells with a TNF-$\alpha$ background (right).

### A.6 BIOLOGICAL RELATIONSHIP RECALL

A valuable use of large-scale HCS experiments is to perform large-scale inference of biological relationships between genetic perturbations. We evaluate each model's ability to recall known relationships by using the *biological relationship recall* benchmark described in Celik et al. (2024). First, we correct for batch effects using *Typical Variation Normalization* (TVN) (Ando et al., 2017), and also correct for possible chromosome arm biases known to exist in CRISPR-Cas9 HCS data

(Lazar et al., 2024). To infer biological relationships, we compute the aggregate embedding of each perturbation by taking the spherical mean over its replicate embeddings across experiments. We use the cosine similarity of a pair of perturbation representations as the relationship metric, setting the origin of the space to the mean of negative controls. We compare these similarities with the relationships found in the following public databases: CORUM (Giurgiu et al., 2019), hu.MAP (Drew et al., 2017), Reactome (Gillespie et al., 2021), and StringDB Szklarczyk et al. (2020) (with >95% combined score). Table 1 reports the recall of known relationships amongst the top and bottom 5% of all cosine similarities between CRISPR knockout representations in RxRx3 (Fay et al., 2023).

## A.7 Replicate Consistency

In order to assess the reproducibility of the perturbations across their technical replicates, we compare the distributions of the similarities for same perturbations across replicates against an empirical null distribution. Specifically, for technical replicate experiments $e_a^i$ and $e_b^i$, we calculate the cosine similarity between the embeddings of perturbation $x_j$ in them, denoted as $s^{x_j}$. The query distribution $q^{e_i}$ is constructed by computing the cosine similarities for all perturbations that have a matching well on experiments $e_a^i$ and $e_b^i$. An empirical null distribution of identical cardinality is created by computing cosine similarity, $r^{x_k,x_l}$, between random pairs from $e_a^i$ and $e_b^i$ such that no pair corresponds to the same perturbation, $p_0^{e_i}$. Using non-parametric statistical tests, namely Kolmogorov-Smirnov (KS) and Cramer Von-Mises (CVM), we can evaluate the hypothesis that $q^{e_i}$ and $p_0^{e_i}$ are drawn from the same distribution. Formally, let $Q^{e_i}(x)$ and $P_0^{e_i}(x)$ be the cumulative distribution functions for $q^{e_i}$ and $p_0^{e_i}$ respectively, then the KS statistic for the two-sample case of technical replicate experiments $e_a^i$ and $e_b^i$ is defined as:

$$\text{KS}^{e_i} = \sup_x |Q^{e_i}(x) - P_0^{e_i}(x)|. \tag{4}$$

The Cramér–von Mises test statistic (CVM) for experiments $e_a^i$ and $e_b^i$ is computed as:

$$\text{CVM}^{e_i} = \frac{1}{2N^2} \sum_{m=1}^{N} \left[ (r_m - m)^2 + (s_m - m)^2 \right] - \frac{4N^2 - 1}{12N}. \tag{5}$$

where $N$ is the cardinality of $q^{e_i}$ and $p_0^{e_i}$ and $s_m$ and $r_m$ are ranks of similarities $s^{x_j}$ and $r^{x_k,x_l}$ in the combined distribution of $q^{e_i}$ and $p_0^{e_i}$ when ordered. In order compare models, we use the median of $\text{CVM}^{e_i}$ and $\text{KS}^{e_i}$ over all technical replicate experiment pairs $e_i$.

Since the pairs are randomly selected for $p_0^{e_i}$, the embeddings would be mostly orthogonal thus the distribution would be centered around 0. Given that not all CRISPR knockouts would induce a morphological change in the cells, it's plausible for distribution $q^{e_i}$ to exhibit a peak around 0. As the model approaches the precision of an oracle, we would anticipate the mass situated around this peak to shift towards higher cosine similarity values.

## A.8 Anax Group Prediction Details

The Anax probing task introduced in this paper is intended to balance capturing a diverse range of biology that is broadly conserved between cell types with a reduced cost of execution. The name "Anax" is a reference to Anaximander, the 6th century B.C. philosopher credited with making the first world map.

In curating these genes, we analyzed the sources listed in § A.6 as well as internal gene expression data to produce "functional" groups corresponding to biological processes, cellular components, and molecular functions. Not all genes within each group are expected to have the same knockout phenotype, but are classified by humans as having related function – linear separability of these genes would indicate that a model has learned similar concepts to those deemed significant by biologists.

The gene groups we use for the 40-class Anax group classification task (§ A.4) are listed in Table 7.

## A.9 Dino-V2 pretraining on microscopy data

We attempted to train two Dino-v2 models on microscopy data. One ViT-L/16 from scratch on RxRx3, and another attempt of fine-tuning the MAE-L/8 on RPI-93M with the Dino-v2 losses.

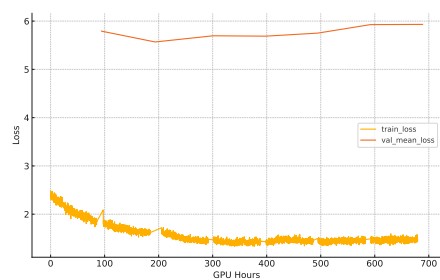

Figure 8: Loss curve when training Dino-v2 ViT-L/16 on RxRx3.

| DinoV2 model | Avg. Prec. | Comp. Z-score | Energy dist. | Energy Z-score |
|---|---|---|---|---|
| ViT-L/16, RxRx3 | $0.258 \pm .015$ | 2.13 | $4.818 \pm 1.49$ | 2.740 |
| ViT-L/8 (ft.) RPI-93M | $0.255 \pm .018$ | 1.76 | $2.705 \pm 0.75$ | 2.626 |

Table 6: RxRx3-core benchmarks for our initial attempts to train Dino-V2 models on microscopy data. The latter was finetuned from the MAE-L/8 trained on RPI-93M. Results compare to Table 3.

They had the following hyperparameter settings which were tuned on another dataset: output-dim 65536, 2 global crops, 4 local crops, dino loss weight 1.0, koleo loss weight 0.1, ibot loss weight 1.0, Lion optimizer with max learning rate 1e-5, weight decay 0.05, betas 0.9 0.95, and cosine annealing. In both cases, we observed significant over-fitting of the loss from the start (Figure 8).

In Table 6 we show that both models fail to improve on the RxRx3-core benchmark metrics (i.e., z-scores over the random baseline) versus the CA-MAE-S/16 RxRx3 model which had Z-score of 2.90 on average precision for predicting compound activity and 3.10 on energy distance between perturbations and controls. We have not found an effective recipe for training Dino on microscopy data. However, we note that theoretical evidence exists arguing that MAE learning is in some ways equivalent to contrastive learning Hondru et al. (2024), so even if an appropriate Dino recipe is found it would remain to be seen if it differs substantially from MAEs for microscopy given the same training compute. As described in the **Limitations** section, we expect that future work would have to dedicate significant training ablations and creativity to determine the best possible training recipe for training Dino on microscopy data.

### A.10 CORRELATION BETWEEN MODEL SCALE AND BENCHMARK RESULTS

In Figure 9 we show the correlations between training FLOps (floating point operations) and downstream results. Over all benchmarks we observe a very strong consistent linear trend where scaling training FLOps improves overall pwerformance. This work provides the next log step in scale as we enter into the billion-parameter model regime with MAE-G/8. These results therefore provide additional evidence that the trend initially discovered by Kraus et al. (2023) between FLOps and relationship recall actually extends both to billion-parameter models and even moreso for other biologically meaningful benchmarks pertaining to linear probes on small experiments and to replicate consistency on the whole-genome.

### A.11 PERFORMANCE ON NEW DATA (JUMP-CP)

In order to validate that the MAEs generalize to entirely novel data, we evaluated a subset of models on completely external public data generated by different assays and from a variety of different labs as produced by the JUMP-CP consortium (Chandrasekaran et al., 2023). Table 2 presents these results using the relationship recall benchmarks of (Celik et al., 2024), noting that only a subset of 7,976 gene-knockouts are covered by this dataset. For post-processing embedding alignment, we use PCA with center-scaling. We observe that the MAEs perform better than the Cellprofiler manual feature extraction baseline (Carpenter et al., 2006b), and that the general trend is maintained with the

Table 7: Anax groups and their associated genes. This table presents a comprehensive list of gene groups and their corresponding genes.

| Anax Group | Genes |
|---|---|
| Acyl Coa Biosynthesis | ELOVL2, ELOVL5, ELOVL6, HACD1, HACD2, HSD17B12, SCD, SCD5, TECR |
| Adherens Junctions | ACTB, ACTG1, AFDN, CDH1, CTNNA1, CTNNB1, CTNND1, NECTIN1, NECTIN3, NECTIN4 |
| Amino Acid Metabolism | ALDH4A1, ARG2, CKB, CKMT2, CPS1, DAO, OTC, PYCR2, PYCR3, SAT1 |
| Apoptosis | CFLAR, DFFB, CASP6, CASP3, FASLG, BCL2, DFFA, XIAP, TNFSF10, AKT3 |
| Autophagy | ATG12, ATG3, ATG4B, ATG4C, ATG7, GABARAP, PIK3C3, PIK3R4, PRKAA1, ULK1 |
| Beta Oxidation Of Fatty Acids | ACAA2, ACADL, ACADM, ACADS, ACADVL, ECHS1, ECI1, HADH, HADHA, HADHB |
| Calcium Signaling | ADCY1, ADCY2, ADCY3, CALM1, CAMK2B, CAMK2D, PDE1B, PDE1C, PRKACG, PRKX |
| Clathrin Coated Vesicles | AP2A1, AP2A2, AP2B1, AP2M1, AP2S1 |
| COPI | ARCN1, COPA, COPB1, COPB2, COPE, COPG1, COPZ1 |
| COPII Vesicles | SEC13, SEC23A, SEC24B, SEC24D, SEC31A |
| DNA Damage Repair | BLM, BRCA2, EME1, NBN, POLD2, RAD51B, RAD51C, RAD51D, RPA1, XRCC2 |
| Dynein | DYNC1H1, DYNC1I2, DYNC1LI1, DYNC1LI2, DYNLT1 |
| ER Protein Translocation | SPCS3, SEC61A1, SRP14, SRP72, SPCS1, SRPRA, SEC11A, SRP68, SRPRB, SRP54 |
| Exosome | DIS3, EXOSC10, EXOSC3, EXOSC4, EXOSC5, EXOSC6, EXOSC7, EXOSC8, EXOSC9, MPHOSPH6 |
| Gap Junctions | ADCY8, DRD2, HTR2C, ITPR2, LPAR1, PDGFD, PDGFRB, PLCB3, TUBA1C, TUBB1 |
| Golgi | ACTR10, ACTR1A, CAPZA3, COG4, CTSZ, PPP6C, RAB1B, SEC22C, SEC24C, TMED9 |
| MAPK | DUSP4, EGF, FGF18, FGF20, HSPB1, MAP2K2, MAPKAPK5, RAC1, RAP1A, RASGRP3 |
| Mitochondria Structure | APOOL, APOO, TMEM11, CHCHD6, ATP5ME, MICOS13, ATP5F1C, DNAJC11, DMAC2L, ATP5MF |
| Mitochondrial Transport | ATP5F1A, COA4, COA6, COX17, HSPA9, IDH3G, PITRM1, PMPCA, PMPCB, SLC25A4 |
| mTOR Pathway | CAB39, CAB39L, EIF4EBP1, MLST8, PRKAA2, RPS6KB1, RPTOR, STK11, STRADA, TSC1 |
| Nonsense Mediated Decay | CASC3, EIF4A3, MAGOH, MAGOHB, RBM8A |
| Nuclear Pore | NUP107, NUP133, NUP153, NUP188, NUP205, NUP37, NUP85, NUP93 |
| Nucleolus Structure | FBL, NAT10, NOLC1, NOP58, UTP20 |
| Nucleotide Metabolism | ADSL, ADSS1, ADSS2, ATIC, GMPS, IMPDH1, IMPDH2, PAICS, PFAS, PPAT |
| P53 Stress Signaling | ATM, ATR, CCNG1, CDK1, CHEK1, CHEK2, MDM2, MDM4, TP53, TP73 |
| Pentose Phosphate Pathway | G6PD, TALDO1, DERA, RPE, PGM2, RBKS, PGD, PGLS, RPEL1, PRPS2 |
| Peroxisome Biology | ACOT8, AGPS, BAAT, HMGCL, HSD17B4, MLYCD, PAOX, PEX12, PEX6, PIPOX |
| Prespliceosome Complex | ALYREF, AQR, CRNKL1, DDX5, HNRNPK, LSM2, PLRG1, PRPF4, SMNDC1, SRSF4 |
| Proteasome | PSMA1, PSMA4, PSMB1, PSMB2, PSMB7, PSMA6, PSMA3, PSMB4, PSMA5, PSMB3 |
| Ribosome Large | RPL13A, RPL11, RPL10, RPL23A, RPL30, RPL7A, RPLP2, RPL28, RPL5, RPL27A |
| Ribosome Small | RPS2, RPS6, RPS8, RPS16, RPS11, RPS3A, RPS19, RPS15, RPS4X, RPS9 |
| RNA Polymerase II | POLR2A, POLR2B, POLR2C, POLR2G, POLR2I, POLR2L |
| TCA Cycle | ACO2, DLST, FH, IDH2, IDH3B, MDH2, OGDH, SDHB, SUCLA2, SUCLG2 |
| Tight Junctions | CLDN14, CLDN17, CLDN18, CLDN19, CLDN4, CLDN8, CLDN9, MPP5, PARD6B, PRKCI |
| Translation Initiation Complex | EIF3G, EIF3A, EIF3D, EIF3I, EIF3K, EIF3M, EIF3B, EIF3H, EIF3E, EIF3L |
| Transport Of Fatty Acids | APOD, LCN12, LCN15, LCN9, SLC27A1, SLC27A4, SLC27A6 |
| Tubulin | TUBA3C, TBCC, TBCD, TUBA4B, TUBA8, TUBAL3, TUBA1A, TUBB4B, ARL2, TUBA1B |
| Unfolded Protein Response | CXXC1, DNAJB11, EIF2S3, KHSRP, MBTPS1, SHC1, TATDN2, TLN1, TSPYL2, YIF1A |
| V-ATPase | ATP6V1A, ATP6V, ATP6V1D, ATP6V1E1, ATP6V1F, ATP6V1H |

trimmed MAE-G/8 obtaining the best recall overall. Notably, recall on JUMP-CP is considerably lower than on RxRx3 (Table 1) likely due to different assay protocols and more variance in the data.

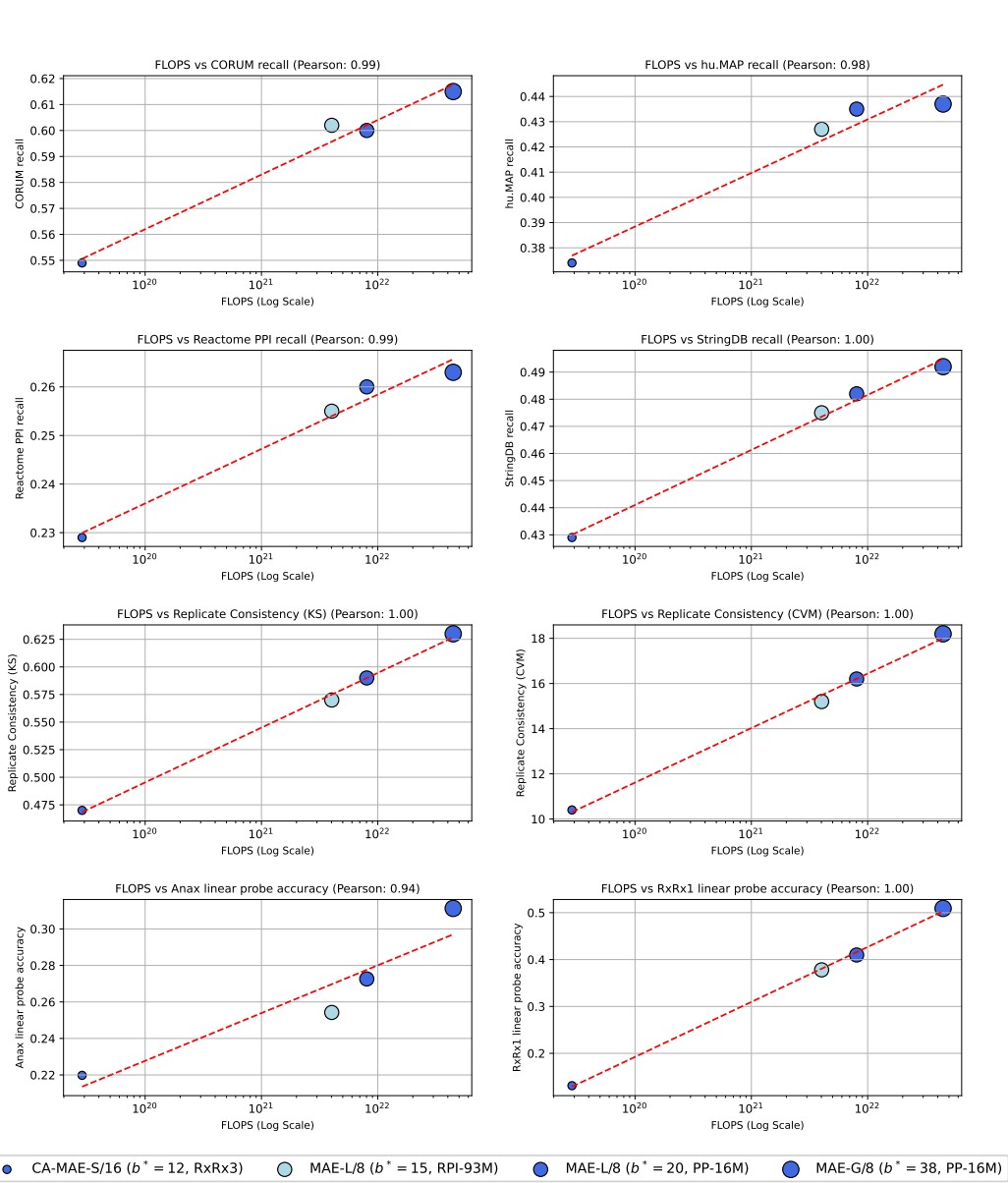

Figure 9: Relationship between FLOPs and benchmark evaluation results for the six whole-genome tasks (Table 1) and the two linear probing tasks (Figure 3).

