# OpenReview forum: "ViTally Consistent: Scaling Biological Representation Learning for Cell Microscopy"
_ICLR.cc/2025/Conference — Submitted to ICLR 2025_

### Official Review · Reviewer_noDg · 2024-10-29

**Soundness:** 1
**Presentation:** 4
**Contribution:** 1
**Rating:** 5
**Confidence:** 4

**Summary:**

The study presents a novel dataset specifically designed to train foundational models for the Cell Painting assay, a high-content imaging technique used for profiling cellular phenotypes. By curating this dataset, the authors aim to address the unique challenges associated with cellular imaging, such as variations in cellular morphology and staining patterns, which require specialized data to train robust and generalizable models.

In addition to introducing the dataset, the study evaluates the effectiveness of several ImageNet-pretrained models in encoding cellular images from the Cell Painting assay. By leveraging pretrained models originally trained on ImageNet, the authors explore how well these architectures can transfer to the domain of cellular imaging.

**Strengths:**

This paper provides an analysis of the critical role dataset curation plays in achieving high-quality representations in microscopy imaging, including the effects of aggressive curation strategies. The work demonstrates how curatorial decisions impact the fidelity and accuracy of data representation.

The authors have presented their findings in a well-organized and accessible manner, with each section building logically on the last. The clarity of the language and structure ensures that concepts are conveyed effectively.

One of the strengths of this paper lies in its visual aids. The figures and images are thoughtfully designed to illustrate core concepts and provide visual clarity, enhancing the reader's understanding of the methodology and outcomes.

The experimental results are convincing to some extent.

**Weaknesses:**

Firstly, I have some reservations regarding the model's evaluation protocol. I recommend that the authors consider using a phenotypic screening benchmark [1]. Specifically, for chemical perturbations, the validation splits should ideally be based on the scaffold, ensuring that drugs with similar structures do not appear in both the training and testing sets. Furthermore, an independent dataset is used to validate the method, which would help clearly assess the influence of batch effects. As an independent dataset I would propose to similar approach as in [1], namely, I would use the trained model and do the linear probing for mode of action prediction.

In terms of contributions, the authors emphasize the importance of data quality. This includes aspects like the overall integrity of the data [2] as well as challenges related to imbalance and undersampling [3]. While these are critical considerations in vision or general datasets, they also apply to microscopy, so the contribution may not be particularly impactful for the research community. I do not see that those issues are specific for microscopy only, they are general for computer vision and transferable to microscopy. The microscopy-related issue is batch effects, and there is not much of their influence on the work. That is why I believe this work's novelty is extremely limited.

Additionally, it appears that the authors have not made the pretrained model or dataset publicly available. While sharing these resources is not a requirement, their absence limits reproducibility, especially given the lack of detailed descriptions in the paper. Without these, reproducing the model’s training process—such as the steps required to train MAE on this dataset—is challenging and limits the model's utility as a foundational resource for the scientific community. I would envision seeing all the details about the quality control step. The descriptions of Appendix A are too general and to make it reproducible, they should be much more detailed. Probably a few times longer description than it is right now.

There are also inconsistencies in the comparison of models. The models were trained with different hyperparameters, which raises questions about the validity of direct comparisons. Lastly, the results in Table 2 lack confidence intervals, making it difficult to assess whether the reported differences are statistically significant. Please add confidence intervals in the results. Additionally, would be great to see if the model trained with the same hyperparameters performs differently.

[1] Borowa, Adriana, et al. "Decoding phenotypic screening: A comparative analysis of image representations." Computational and Structural Biotechnology Journal 23 (2024): 1181-1188.

[2] Cole, Elijah, et al. "When does contrastive visual representation learning work?." Proceedings of the IEEE/CVF Conference on Computer Vision and Pattern Recognition. 2022.

[3] Johnson, Justin M., and Taghi M. Khoshgoftaar. "Survey on deep learning with class imbalance." Journal of big data 6.1 (2019): 1-54.

**Questions:**

I do not have questions about the paper per se. I do not see novel and reproducible enough to be published.

---

> ### Author Response · Authors · 2024-11-22
>
> ## Batch effect
> > The microscopy-related issue is batch effects, and there is not much of their influence on the work.
>
> In this work evaluate these models on a benchmarking task called “Replicate Consistency”, which has not been studied before for this class of models. This benchmark is precisely designed to measure the model’s ability to handle batch effects - it measures the degree to which the model produces similar embeddings for images of the same perturbations from different experimental batches. MAE-G/8 obtains the best performance on this task by a significant amount as shown in our general reply above. This result indicates that scaling MAE on curated data not only yields improvements in biological relationship recall, but also in batch correction.
>
> ## Evaluation
> Regarding the evaluation protocol: note that the models are not provided perturbation information (gene/guide identity or chemical structure), neither in training nor at evaluation time; they are fully self-supervised on the image data alone. Furthermore, the majority of the relationship tasks we evaluate (CORUM, hu.MAP, Reactome, StringDB) are gene-gene benchmarks, which do not implicate chemical perturbations at all, making scaffold splits out of scope. We have benchmarked our work on the RxRx3 and RxRx3-core datasets, which are single-source, many-batch datasets enabling us to test the effectiveness of these models in a setting with real-world levels of batch-to-batch variance. Finally, while we do not implement the suggested Borowa 2024 benchmark, in the revision we have added the similar RxRx3-core benchmark, in which we measure zero-shot target prediction performance via cosine similarities.
>
> Both benchmarks in the paragraphs above were described in detail in the Celik et al. PLOS Comp Bio article, the contents of which have been on bioarxiv since December 2022. These benchmarks are well validated on a variety of datasets for use cases relating perturbations like the ones we describe in our work. The benchmark proposed by noDg (a five layer neural network trained to predict 30 mode of action classes and 121 property prediction tasks) is in effect similar to the Anax linear probing tasks which we present here, using classes applicable to gene knock-outs. The biological relationship recall task we use is based on direct comparisons of MAE features across every perturbation in a genome wide knock-out screen (~17,000x17,000) compared to ~65,000 annotated relationships to compute relationship recall. To demonstrate that our models generalize to out of distribution data, we also report the biological recall metric on JUMP-CP data, which we have added to the appendix.
>
> ## Hyperparameters
> We will add more detailed description of the data curation steps in the final version of the paper (the co-author who worked on the curation step is currently on vacation).
>
> ViTs are trained with different hyperparameters because differently sized models require different hyperparameters to train efficiently and avoid divergence. We do not possess the computational resources required to ablate every hyperparameter when training these large models. Instead, we tuned hyperparameters early on in initial training runs until evidence suggested good dynamics, as well as basing them on hyperparameter choices from related work. We pursued ablations on the two primary hyperparameters in this work: (1) the training dataset; (2) the model scale. We only had the compute budget to train one complete iteration of the MAE-G/8, so we had to bet on what would be the most likely recipe for success, which is why we trained it on Phenoprints-16M as this worked well for the MAE-L/8 backbone.
>
> Finally, we have added additional significant figures and indicated the standard deviation for the results in the main table (including Table 2) by clarifying the caption.

---

> > ### Comment · Reviewer_noDg · 2024-12-01
> >
> > Batch Effects
> > Given the existence of established benchmarks for studying batch effects, the first step should be to provide results using these existing benchmarks. If a new benchmark is proposed, it must be part of a well-justified contribution that includes a detailed analysis explaining why this new benchmark is necessary, how it differs from, and why it is superior to, the currently used benchmarks. Demonstrating results on existing benchmarks would strengthen the work and lay the groundwork for introducing a novel benchmark later.
> >
> > Thank you for clarifying the evaluation strategy. I would also like to inquire about the differences observed between gene-based and chemical-based benchmarking. A discussion of these differences would be valuable in understanding the broader implications of the work.
> >
> > Hyperparameters and Reproducibility
> >
> >
> > It is unfortunate that some team members are unavailable due to vacations. However, as a reviewer, I am obligated to base my assessment on the information provided in the revised manuscript and accompanying discussion. Since specific details are not included, I cannot assume they will be provided.
> >
> > Given the revisions and clarifications provided, I will raise my score from 1 to 3.

---

> > > ### Author Response · Authors · 2024-12-02
> > >
> > > Thank you for your response. We do use established benchmarks for studying batch effects in this work - that is precisely why we use the RxRx1 task as a linear probe. Its 2023 paper is well-cited and is titled "Rxrx1: A dataset for evaluating experimental batch correction methods". Furthermore, our replicate consistency benchmark is designed based on the previously mentioned established benchmarks proposed by Celik et al.; we provide a detailed explanation, motivation, and analysis of this benchmark (047-051, 365-369, and in the Appendix 927-954). If you are referring to the benchmarks in the aforementioned paper "Decoding phenotypic screening: A comparative analysis of image representations", it is essentially contemporaneous as it is from 2024 (published in PubMed in March and Elsevier in December), whereas the previously mentioned works have been online longer and are more well-known - we appreciate the insights from that work and we will add a citation to it in the final version of the paper.
> > >
> > > We believe that our results are important and valuable for the scientific community as they demonstrate that our proposed methods, including scaling MAE, naturally yields representations that offer significant improvements in correcting for batch effects.
> > >
> > > Regarding your inquiry, one of the important differences between gene-based and chemical-based benchmarking in high-content assays comes from the differences in data background. That is, gene-knockout perturbations use CRISPR which may induce DNA damage response in the cells, whereas chemical perturbations require solvents such as DMSO to induce the perturbation into the cells. Future work could therefore pursue new approaches for alignment, building off and improving upon the zero-shot gene-activity prediction results we present in this work. Nonetheless, we observe that our proposed scaling approach yields consistent improvements on this task as well.

---

> > > > ### Comment · Reviewer_noDg · 2024-12-03
> > > >
> > > > I've changed my score to 5, however, I still think that the current version is not yet ready to be published due to the limited novelty (I am still not convinced).

---

### Official Review · Reviewer_8oYL · 2024-11-02

**Soundness:** 3
**Presentation:** 4
**Contribution:** 3
**Rating:** 8
**Confidence:** 3

**Summary:**

The authors:

Curate a large dataset (Phenoprint-16M) using a unique combination of previously introduced curation methods.
Introduce a ViT-G/8 MAE foundation model for cell microscopy imaging which outperforms the previous STOA model.
Show, using linear-probing, that mid-level representations from their model can outperform the representations from deeper layers enabling cheaper inference and improved performance.
Show that scaling to a 1.9 billion parameter MAE model continues to improve the performance for cell microscopy imaging / HCS.

**Strengths:**

The paper is very well written and provides clear explanations of the experiments and dataset curation. I believe that this will help future researchers better curate their own datasets.
Their foundation model outperforms the previous SOTA and would be valuable to the scientific community if made publicly available.
The required effort to produce the curated dataset and training of the large-scale model was substantial.
Although it has been shown in other domains, the evidence that mid-level representations can be of higher quality than deeper layers is valuable for this field as it reduces the cost of inference and potential fine-tuning.

**Weaknesses:**

It’s already well-established that scaling up models + dataset curation helps performance, these findings are not novel.
Only one method to evaluate the quality of the representations is used (linear-probing). It would be more convincing if other measures were given as well (kNN, fine-tuning). Fine-tuning evaluations on down-stream tasks using a varying number of blocks would have been useful as well. I understand that this may have been too expensive for all layers, but could have been done for a select number of layers.
No comparison is made between DINO-V2 pretrained on microscopy images versus MAE pretrained on microscopy images. Are the MAE results better because of the training method or just the training data? There’s nothing in this paper that convinces me that an MAE is more suitable than any other SSL techniques.
Why is Figure 5 comparing the ViT-G/8 MAE model against the DINO-V2 G/14 pretrained on natural images? The results in Table 2 already show that pretraining on microscopy images is critical. It feels like an unnecessary comparison. I believe a more fair comparison would be against MAE-L/8 , RPI-93M, the previous STOA.

**Questions:**

Will the ViT-G/8 MAE foundation model be made publicly available? I see no mention of this.
The bolding in table 2 is inconsistent.

---

> ### Author Response · Authors · 2024-11-22
>
> Please see the general reply which addresses most of your questions. We originally had Figure 5 comparing the natural image Dino-G to the MAE-G in order to demonstrate the difference between using an off-the-shelf natural image model in its standard form (untrimmed) compared to the contribution of our paper (using the new model trimmed). We appreciate your feedback and have decided to remove this figure from the paper. In the updated manuscript we have added a new figure (Fig. 6) visualizing the Energy distance distributions from 3 MAEs on the public RxRx3-core benchmark.

---

> > ### Author Response · Authors · 2024-12-01
> >
> > As the review period comes to an end we would like to please ask if our replies and the updated manuscript have addressed your comments and if there are any additional comments you like to make?

---

### Official Review · Reviewer_bwiB · 2024-11-04

**Soundness:** 2
**Presentation:** 3
**Contribution:** 2
**Rating:** 5
**Confidence:** 5

**Summary:**

In this work, the authors present a large 1.9B parameter masked auto-encoder model (MAE-G/8) trained on over 8 billion microscopy image crops. In addition to scaling dataset and model size, they also explore data curation strategies and evaluation on proxy tasks for searching for the optimal representation for biologically relevant tasks from the transformer architecture. They introduce a data curation strategy that filters perturbations that do not have a statistically significant consistent morphological change in cells compared to negative controls on smaller pre-trained networks (WSL and MAE-L/8).

Through their experiments, they demonstrate that intermediate layer representations capture 60% more linearly separable latent space which is also more biologically informative (improvements in both recall and replicate consistency) across all models pre-trained on ImageNet and models trained on microscopy datasets. The paper also presents a manually curated 40-class dataset named Anax and compare performance of their proposed methods against baselines across Anax, RxRx1 and whole genome dataset.

**Strengths:**

* This paper explores the following important aspects of training and scaling large foundational representation models for microscopy images such as
  - dataset curation,
  - proxy linear probing tasks for reasonable evaluation of models during training that generalizes to downstream performance and
  - selection of optimal intermediate layer representation for specific downstream tasks
* The paper shares details of a manually curated dataset of 40 functionally-diverse gene groups containing 348 genes which provide a useful dataset for linear probing proxy tasks and evaluation of the models

**Weaknesses:**

1. While the paper shows performance improvements with model scale on microscopy images, the overall contribution is incremental and lacks sufficient ablation studies.
2. While the authors hypothesize why their dataset curation might be better compared to other dataset curation strategies, there is no experimental comparison of different data curation strategies on microscopy datasets
3. There is very little novel contribution in the selection of optimal block for specific downstream task based on balanced accuracy on linear classification task
4. The gains in biological recall metric (across all 4 gene network databases) for the larger MAE-G/8 seem very modest compared to the MAE-L/8 baselines (Table 2).
5. The biological recall metrics for trimmed vs untrimmed versions of the MAE models trained on microscopy images are mostly the same which is counterintuitive to the argument that intermediate block representations are more useful.
6. The paper describes a data curation and a methodology for trimming blocks of ViT, it specifically explores the two options for training MAE models. While DINO-ViTs trained on ImageNet are a useful reference, additional experiments with DINO-ViT trained on microscopy images would be required to demonstrate the general applicability of the proposed data curation and ViT trimming strategies.
7. The paper does not explicitly state that the dataset and model weights will be available to the public. Metrics shared on private datasets (manually curated datasets) are insufficient to use for evaluation of scientific contribution.
8. While CA MAE was used as a baseline, the results and lower performance of CA-MAE compared to other baselines is not discussed.

**Questions:**

* Dataset Availability: Will the 384-gene manually curated Anax dataset and curated Phenoprints-16M datasets be released as part of this submission?
* Model Availability: Will the ViT-G/8 model weights be released as part of this submission?

---

> ### Author Response · Authors · 2024-11-22
>
> Please see the general reply which addresses most of your comments.
>
> Using the complete Anax gene group we provided in Table 7, the majority of the Anax curated dataset can be easily obtained from the publicly available gene-KO labels in the public RxRx3 dataset, or from the RxRx3-core dataset.
>
> Ablation/comparisons: while testing all 8 (2×2×2) combinations of our key components—block search, data curation, and model scaling—would be ideal, it is prohibitively expensive, requiring more than four times the compute cost of our current study. However, we included targeted comparisons to isolate the impact of each component (Table 1, Figure 1B). Specifically, Figure 1B compares (1) block search (bar 1 vs. 2), (2) training data (bar 2 vs. 3), and (3) model scaling (bar 3 vs. 4), while holding the other two factors constant. The final version, combining all components, achieved the best performance.

---

> > ### Author Response · Authors · 2024-12-01
> >
> > As the review period comes to an end we would like to please ask if our replies and the updated manuscript have addressed your comments and if there are any additional comments you like to make?

---

### Official Review · Reviewer_Y4dq · 2024-11-04

**Soundness:** 3
**Presentation:** 3
**Contribution:** 2
**Rating:** 3
**Confidence:** 4

**Summary:**

The authors presented a new 1.9 billion-parameter ViT-G/8 MAE model, trained on over 8 billion microscopy image crops. The authors demonstrated performance boost over a smaller ViT-L/8 model in various downstream tasks.

**Strengths:**

- Cell microscopy analysis is an important problem and has a large impact on the future of computational pathology and bioinformatics research.
- The authors curated a dataset to only include relevant data using biological informed methods, which may be a worthwhile contribution.
- The authors performed careful tuning of model architectural and training details.

**Weaknesses:**

- The methodological contribution is limited. The authors applies MAE, and evaluated on various downstream tasks. It is also known that newer algorithms such as iBOT or DINOv2 that are more effective in learning visual representations. The authors should benchmark these methods as well.
- The author’s model performance boost over a much smaller MAE-L/8 trained on RPI-93M model is very marginal, and the benchmarks do not have error bars.

**Questions:**

- Please address the weakness mentioned above, especially regarding the model evaluation and model training algorithm.
- Are the data, model, and training code going to be publicly available?

---

> ### Author Response · Authors · 2024-11-22
>
> Please see the general reply regarding methodological contribution and the results.
>
> We have added additional significant figures and indicated the standard deviation for the results in the main table by clarifying the caption.
>
> All model training code and the full weights for CA-MAE will be released publicly - we indicate this after the conclusion of the updated manuscript.

---

> > ### Author Response · Authors · 2024-12-01
> >
> > As the review period comes to an end we would like to please ask if our replies and the updated manuscript have addressed your comments and if there are any additional comments you like to make?

---

### Author Response · Authors · 2024-11-22
**General reply**

Thank you to all the reviewers for your thoughtful comments. First, we reiterate the main takeaway of our paper: combining three strategies to generate representations of cell microscopy images that address a key challenge in image-based drug discovery—achieving replicate consistency while preserving strong biological recall.

# Reproducibility

We updated our work to address major critiques regarding reproducibility. Specifically, in addition to the linear probing results presented on the public RxRx1 dataset, we added results on a new public benchmarking dataset, RxRx3-core, for the primary models under investigation. As for data curation, unfortunately the author responsible is not available right now, but we will be sure to add ample details in the final version once he returns.

Combined with our commitment to publicly release the code and full weights for CA-MAE-S/16, we believe that these results strengthen our paper and enable better reproducibility. Phenoprints-16M is a >1PB dataset, larger than the already publicly available 200TB RxRx3 dataset, which itself has been understudied. While at this time we cannot release this new dataset and MAE-G/8 weights, we believe that it is still valuable to publicly report these results (which offer evidence into what works effectively for industrialized drug discovery applications) rather than keeping these results private.

# Methods

We agree that it would be valuable to appraise different learning algorithms on this data to try improving on MAE. We pursued a variety of more “novel” techniques during development; e.g., we spent thousands of GPU hours attempting supervised contrastive learning (which had poor performance so we did not include their results).

We also experimented with training Dino-V2 on microscopy data, and include results in the Appendix. We were unable to prevent Dino from overfitting on batch effect features (the validation set contains fully held-out batches during training); in contrast, MAE scaled to billions of parameters with strong generalization to reconstruct held out batches from the validation set. We look forward to future work to discover a performant bespoke recipe for applying Dino-V2 or other SSL techniques on large volumes of microscopy data.

# Novelty

As for novelty, we combine three well-known methods for effective representation learning: curating the training data, linear probing intermediate representations, and scaling the model. This is why the primary area of this paper is “Applications” and not “general representation learning” or “computer vision”.

Our curation and probing techniques are novel in their practical application to this context; for example, our curation strategy leverages the experimental data generation process to identify samples with features significantly different from controls. Additionally, we demonstrate that features learned from the MAE loss can be directly applied to meaningful use cases in biological research and drug discovery without additional fine-tuning. Finally, the Anax linear probing task required significant biological expertise to articulate the classification groups, and this classification task can now be used more broadly even beyond imaging (e.g. for evaluating models of gene-knockouts in single-cell RNAseq datasets).

# Results

While our findings may not be surprising to every reader, we certainly did not take for granted that decreasing the training dataset size by a factor of nearly 6x (93M → 16M) while increasing the model parameters by a factor of nearly 6x (330M → 1.86B) would yield improved results. We believe this result is valuable in itself, as it indicates that improvements for MAE on this data are more likely to be obtained by increasing the model scale given these dataset sizes.

Summarizing key performance improvements offered by MAE-G/8 trimmed over the previous SOTA (MAE-L/8 final layer, trained on RPI-93M):

Relationship recall (noting each database comprises between 15,000 to 46,000 relationship pairs):
- CORUM: +1% = 175 more relationships recalled
- hu.MAP: +0.7% = 138 more relationships
- Reactome-PPI: +3% = 187 more relationships
- StringDB: +4% = 884 more relationships

Replicate consistency (i.e., ability to create consistent representations across batches):
- KS: +21%
- CVM: +48%

Linear probing (noting that the validation set is held out experimental batches):
- RxRx1: +60%
- Anax: +38%

RxRx3-core (publicly available benchmark dataset for zero-shot evaluation of embeddings):
- Gene-compound retrieval: +42%
- Perturbation-control energy distance: +17%

Prior work has primarily focused on relationship recall; we demonstrate consistent performance improvements across a diverse set of biologically relevant challenges. These findings were far from predetermined; scaling the model architecture introduced potential risks of tradeoffs in specific tasks. These findings confirm scaling trends in vision on natural images extend to biological imaging.

---

### Meta-Review · Area_Chair_sjVC · 2024-12-23

**Metareview:**

The paper presents a new unsupervised vision transformer foundation model of cell microscopy with the goal of using the model for downstream tasks including predicting the cellular phenotype of chemical and genetic perturbations. The main contributions are scaling model size, curation of the training dataset, and the discovery that intermediate layers of the model are more informative for downstream tasks. The reviewers cite the lack of methodological novelty and the marginal benefit from model scaling to largely recommend rejection.

**Additional Comments On Reviewer Discussion:**

There was limited engagement by reviewers during the discussion period, perhaps due to the poor initial reception of the paper. However, even reviewers who engaged with the authors were dissatisfied with the extent to which the revision addressed their concerns.

---

### Decision · Program_Chairs · 2025-01-22

Reject